# Deficiency of AP1 Complex *Ap1g1* in Zebrafish Model Led to Perturbation of Neurodevelopment, Female and Male Fertility; New Insight to Understand Adaptinopathies

**DOI:** 10.3390/ijms24087108

**Published:** 2023-04-12

**Authors:** Luca Mignani, Nicola Facchinello, Marco Varinelli, Elena Massardi, Natascia Tiso, Cosetta Ravelli, Stefania Mitola, Peter Schu, Eugenio Monti, Dario Finazzi, Giuseppe Borsani, Daniela Zizioli

**Affiliations:** 1Department of Molecular and Translational Medicine, University of Brescia, Viale Europa, 11, 25123 Brescia, Italy; 2Neuroscience Institute, Italian Research Council (CNR), 35131 Padova, Italy; 3Istituto di Ricerche Farmacologiche Mario Negri IRCCS, 24126 Bergamo, Italy; 4Department of Biology, University of Padova, Via Ugo Bassi 58/B, 35131 Padova, Italy; 5CN3 “Sviluppo di Terapia Genica e Farmaci con Tecnologia ad RNA”, 25123 Brescia, Italy; 6Department of Cellular Biochemistry, University Medical Center, Georg-August University, Humboldtallee 23, 37073 Gottingen, Germany; 7Clinical Chemistry Laboratory, ASST Spedali Civili di Brescia, 25123 Brescia, Italy

**Keywords:** intracellular vesicular trafficking, adaptinopathies, embryonic development, CRISPR/Cas9 technique

## Abstract

In vertebrates, two homologous heterotetrameric AP1 complexes regulate the intracellular protein sorting via vesicles. AP-1 complexes are ubiquitously expressed and are composed of four different subunits: γ, β1, μ1 and σ1. Two different complexes are present in eukaryotic cells, AP1G1 (contains γ1 subunit) and AP1G2 (contains γ2 subunit); both are indispensable for development. One additional tissue-specific isoform exists for μ1A, the polarized epithelial cells specific to μ1B; two additional tissue-specific isoforms exist for σ1A: σ1B and σ1C. Both AP1 complexes fulfil specific functions at the *trans*-Golgi network and endosomes. The use of different animal models demonstrated their crucial role in the development of multicellular organisms and the specification of neuronal and epithelial cells. *Ap1g1* (γ1) knockout mice cease development at the blastocyst stage, while *Ap1m1* (μ1A) knockouts cease during mid-organogenesis. A growing number of human diseases have been associated with mutations in genes encoding for the subunits of adaptor protein complexes. Recently, a new class of neurocutaneous and neurometabolic disorders affecting intracellular vesicular traffic have been referred to as adaptinopathies. To better understand the functional role of AP1G1 in adaptinopathies, we generated a zebrafish *ap1g1* knockout using CRISPR/Cas9 genome editing. Zebrafish *ap1g1* knockout embryos cease their development at the blastula stage. Interestingly, heterozygous females and males have reduced fertility and showed morphological alterations in the brain, gonads and intestinal epithelium. An analysis of mRNA profiles of different marker proteins and altered tissue morphologies revealed dysregulated cadherin-mediated cell adhesion. These data demonstrate that the zebrafish model organism enables us to study the molecular details of adaptinopathies and thus also develop treatment strategies.

## 1. Introduction

Proteins and lipids are transported between organelles by vesicles. Several secretory transport routes starting at the trans-Golgi network and endocytic routes are mediated by heterotetrameric adaptor protein (AP) complexes. Eukaryotic cells have five homologous AP-complexes named: AP1, AP2, AP3, AP4 and AP5. Each AP complex has a characteristic intracellular distribution, protein sorting motif-signal recognition, specificity and function. Only AP1 and AP2 complexes form clathrin-coated vesicles (CCVs). Tissue-specific isoforms exist for AP1, AP2 and AP3 in vertebrates [1]. Two ubiquitous complexes, AP1G1 and AP1G2, are formed by the highly homologous γ1 or γ2 subunits (adaptins), respectively, and β1, μ1A and σ1A. The γ-adaptins mediate organelle binding specificity, β1 mediates clathrin binding and μ1A and σ1A mediate the binding of specific vesicle cargo proteins. AP1G1 and AP1G2 are each essential for vertebrate development and cannot compensate for one another. The γ1 and γ2 adaptins, in addition to mediating membrane binding specificity, also recruit via their C-terminal domains co-adaptors and accessory-proteins, which are required for vesicle formation and for the binding of cargo proteins not recognized by μ1A and σ1A subunits. Their sequence differences are concentrated in their C-terminal domain, which indicates that they form different classes of transport vesicles. In addition to these ubiquitous AP1 complexes, some tissues express tissue-specific AP1 isoforms. In these are the ubiquitous μ1A or σ1A adaptins replaced by a highly homologous isoform. The AP1G1 complex with the polarized epithelial cells specific to the μ1B (AP1M2) adaptin mediates basolateral protein recycling [2] and transports transmembrane proteins to somatodendritic domains in neurons [3,4,5]. The two σ1A (AP1S1) isoforms σ1B (AP1S2) and σ1C (AP1S3) have a complementary organ-specific expression pattern so that most tissues express σ1A and only one of its isoforms. Interestingly, AP1G1 complexes have also a function in protein transport independent of CCV formation. They regulate the maturation of early endosomes into late, multivesicular body endosomes. The ubiquitous AP1G1 stimulates maturation by forming an AP1G1 σ1A-ArfGAP1-Rabex-5 complex, which leads to the activation of the Rab5-Vps34 pathway and an increase in phosphatidylinositol-3-phosphate. AP1G1 σ1B reduces the formation of this highly stable membrane-bound complex by binding to Rabex-5 directly. This is the reason for the higher rate of early to late endosome maturation and protein transport in pre-synapses of σ1B knockout mice. The σ1B encoding gene *AP1S2* is encoded on the X-chromosome in mice and man and its deficiency causes severe X-linked intellectual disability and impaired motor coordination [6].

In the past, the generation of knockout and knockdown animal models highlighted the important roles of AP1 subunits in vesicular traffic and their indispensable functions in the development of vertebrates and their various tissues and the immune system [7,8].

It also demonstrated that μ1B can partially substitute μ1A and that also σ1B and σ1C can only partially substitute σ1A (P. Schu, unpublished). However, we do not know all isoform-specific functions of the adaptins and the AP1 complexes they are part of.

Recent research led to the identification of adaptin mutations that reduce adaptin levels and impair functions. These are novel adaptinopathies whose molecular mechanisms are not known. Mice with a spontaneous deletion of two γ1 amino acids are viable, but exhibit multiple phenotypes in the inner ear, retina, thyroid and testes. They also showed circling behaviour and were unable to swim. The multiple pathologies associated with the *AP1G1* hypomorphic mutation may be the result of mislocalized membrane proteins in polarized cells of the affected tissues [9]. Comparable phenotypes were not observed in *AP1G1* heterozygous mice, which indicates that this shortened γ1 mutant may have dominant negative effects only in specific tissues and cell types [8]. The expression of the μ1B isoform is restricted to polarized epithelia. The μ1B KO mice are viable, but have severe defects in gut morphology leading to severe bacterial infections [10,11]. The difference in phenotypes between γ1 and μ1A KO mice suggests that μ1B can also fulfil functions of the ubiquitous μ1A in development. That μ1B can substitute μ1A in lysosomal protein sorting, an AP1G1 housekeeping function, has been demonstrated [12,13,14]. Only recently has it been shown that μ1B is indeed already synthesized at very early developmental stages [15]. Vertebrates do express two σ1A-adaptin tissue-specific isoforms, σ1B and σ1C (AP1S2, AP1S3). Most tissues express only one of these two isoforms besides σ1A. The σ1B KO mouse is viable and fertile but does have severe defects in synaptic vesicle recycling and synaptic protein sorting, leading to impaired learning and memory and disturbed motor coordination. These pre-synaptic phenoytpes are caused by altered AP1G1, not AP1G2, functions [3,16,17]. Additionally, humans deficient in σ1B show severe intellectual disability and impaired motor coordination [18,19]. Biallelic mutations of the AP1S1 gene encoding σ1A cause an autosomal recessive multisystem disorder characterized by mental retardation, enteropathy and peripheral neuropathy, named MEDNIK-syndrome [20,21]. Recently, several point mutations in γ1 have been identified in humans, which develop MEDNIK-like symptoms [22]. Faint expression of a truncated σ1A was detected in patients with MEDNIK syndrome. Other AP1S1 mutations were only detected in heterozygous patients, which indicates that their homozygosity might be embryonic lethal [21]. However, lethality might also be caused by the dominant negative effects of mutated σ1A adaptins. In humans, the recessive loss-of-function of AP1B1 variants (β1) causes a multi-organ disorder with clinical manifestations, including ichthyosis, deafness and photophobia [23]. The presence of two-point mutations in the AP1B1 gene causes keratisichthyosis-deafness and developmental delay, named KIDAR syndrome [24]. Patients lacking in β1 expression develop MEDNIK-like syndromes [25]. Alterations in the sorting of several proteins have been described [26,27]. Recently, 11 families with different mutations in AP1G1 were identified and all individuals show the same clinical features of a neurodevelopmental disorder (NDD) with global developmental delay and intellectual disability (ID), with severity from mild to severe [22] All these adaptinopathies show a prevalence of neurological symptoms, sometimes due to the mutation of a brain-specific isoform. However, neurons are the cells with the most active vesicular protein trafficking essential for neuron functions. This makes these cells especially vulnerable to defects in protein sorting. We do not understand the molecular mechanisms causing the different phenotypes of these adaptinopathies and, therefore, the molecular adaptin functions have to be studied in more detail. To gain more insight into the mechanism of human pathology and functions of the γ1 subunit (Ap1g1), we established an *Ap1g1* (γ1) knockout zebrafish model using the CRISPR/CAS9 technique. Our previous studies on AP1 adaptin functions in zebrafish demonstrated the conservation of vertebrate adaptin functions [16,28,29]. The γ1^−/−^ fish cease development during the transition between the blastula and gastrula stage, in line with the γ1^−/−^ mouse phenotype, which ceases development prior to the blastocyte hatching out of the zona pellucida. Interestingly, adult heterozygous fish showed impaired cell adhesion in epithelial tissues, such as cadherin protein sorting and adherent junction (AJ) formation. Female and male heterozygous animals have reduced fertility due to impaired oocyte and sperm maturation. This analysis revealed novel essential functions of the vertebrate AP1G1 complexes, which cannot be fulfilled by its highly homologous AP1G2 complex. These data demonstrate the usefulness of the zebrafish model systems for future analyses of human adaptinopathies.

## 2. Results

### 2.1. Generation of Ap1g1 Mutant Line

The zebrafish Ap1g1 protein consists of 819 amino acids and shows a high amino acid identity (95%) with the human and mouse counterparts. Zebrafish genome encodes only for one ortholog protein, which has 91% identity with humans and mice; the *ap1g1* transcript is expressed from early to late stages of development, particularly in the developing central nervous system (CNS) and somites [3]. To generate ap1g1 loss-of-function, using zebrafish as a vertebrate model, we mutated the *ap1g1* gene using a CRISPR/Cas9 approach [30]. A suitable single guide RNA (sgRNA) sequence was selected in exon 1 by the CHOPCHOP algorithm (https://chopchop.cbu.uib.no, accessed on 1 March 2015). The sgRNA, injected together with Cas9 mRNA, caused mutation at the expected target site in the *ap1g1* gene in F0 embryos and adults. Out of 200 injected embryos, 136 reached reproductive age and 52 out of 136 were genotyped at 6 weeks by DNA sequence analysis to verify the presence of the mutation (F0). As confirmed by sequencing of PCR fragments, five individuals had a mutation in the proximity of the sgRNA target sequence (Appendix A). Of the 90 analyzed heterozygous F1 offspring animals, 33 were found to carry two types of mutations in exon 1 in the proximity of the PAM site: the first one is an insertion of two nucleotides, the second one is a deletion of two and an insertion of four nucleotides. Both are frameshift mutations leading to premature stop codons. All genotypes could be easily distinguished by PCR combined with high-resolution melting analysis (HRMA) and confirmed by Sanger-sequencing of DNA obtained by fin clips at 6 weeks of age. The heterozygous F1 offspring with the 2-nucleotides insertion were used to obtain the F2 generation (*ap1g1ubx2*-line in ZFIN (database https://zfin.org/ZDB-ALT-230130-5, accessed on 2 April 2023). We decided to carry out our further analysis on individuals with the mutation of two nucleotide insertion lines whose genotypes will hereinafter be referred to as *ap1g1^+/+^*; *ap1g1^+/−^* and a*p1g1*^−/−^. At 4.5 h post-fertilization (hpf), we analyzed using qPCR the expression profile of *ap1g1* mRNA and we observed a significant reduction (>90%) in *ap1g1^−/−^* mutant embryos compared to *ap1g1*^+/+^ and *ap1g1*^+/−^ individuals, suggesting the activation of the nonsense-mediated mRNA decay process (Figure 1D).

### 2.2. Lethality of ap1g1 Mutants during Development

We analyzed the genotypes of F2 progeny at different developmental stages: 3.5 hpf and subsequently at 3-, 28- and 90-days post fertilization (dpf) (Figure 1A). At 3 dpf and later stages, we did not find *ap1g1*^−/−^ animals and we suggested that the lack of *ap1g1* subunit is lethal. In the F2 progeny at 3.5 hpf, it was possible to differentiate two main phenotypes: embryos with normal morphology and embryos with dramatic morphological abnormalities in dividing cells of the yolk syncytial layer. Genotype analysis revealed that embryos with normal morphology were either +/+ or +/− while the aberrant ones were −/−. Therefore, we genotyped fertilized eggs (*n* = 376) of the F2 generation at 3.5 hpf and we found a genotype distribution as follows: 27% *ap1g1*^+/+^ (*n* = 104); 47% of *ap1g1*^+/−^ (*n* = 177) and 26% *ap1g1*^−/−^ (*n* = 95) (Figure 1B). We followed the development of both +/+ or +/− embryos till 120 hpf and we did not detect any macroscopic morphological abnormalities. To verify that the embryos of different genotypes were at the same developmental stage (blastula), we analyzed *desmoplakin b* (*dspb*) expression by whole mount in situ hybridization (WISH). For epidermal morphogenesis, *dspb* is required, and its transcripts are maternally provided at the blastula stage, detectable before zygotic transcription activation, and widely distributed at the gastrula stage [31]. At the blastula stage, both morphologically normal and abnormal embryos expressed *dspb* with the same distribution, indicating the proper embryonic development of *ap1g1*^+/+^, *ap1g1*^+/−^ and impaired *ap1g1*^−/−^ individuals until that developmental stage (Figure 2A). At 4 hpf, we also analyzed the presence of cells death in developing embryos by acridine orange staining and found that cells present in the animal pole of the *ap1g1*^−/−^ embryos undergo cell death compared to *ap1g1*^+/+^ and ap1g1^+/−^ embryos at the same developmental stage (Appendix A). Although in silico analysis of the sgRNA used to generate the *ap1g1* null^−/−^ alleles predicted the absence of off-target sites in the zebrafish genome, we sought to determine the specificity of the blastula phenotype shown by *ap1g1*^−/−^ individuals by injecting ap1g1 wild-type zebrafish mRNA into embryos at the one-cell stage and assessing whether the percentage of embryos survival could be rescued. To validate the cause–effect relationship between the mortality of *ap1g1*^−/−^ embryos and the lack of ap1g1 mRNA, we injected the optimal dose of 100 pg/embryo of zebrafish *ap1g1* mRNA in F2 generation. This significantly rescued the lethality of *ap1g1*^−/−^ embryos and the surviving embryos were able to initiate epiboly and undergo normal embryonic development indistinguishable from wild-type ones. All injected embryos were observed till 48 hpf and then genotyped; the selected amount of *ap1g1* mRNA injected into F2 generation embryos is able to reduce the lethality of 78% (95% mortality vs. 17%) as shown in Figure 2B. The obtained results confirm the specificity of the lethal *ap1g1*^−/−^ embryos. This is in line with our previous studies performed in zebrafish with ap1g1 antisense morpholino and the data from Ap1g1 mouse knockout embryos, which ceased development at the blastocyst stage [2,3].

### 2.3. Reduced Fertility of ap1g1^+/−^ Animals

Adult *ap1g1*^+/−^ fish reached at least 18 mpf (months post-fertilization) as the *ap1g1*^+/+^ counterpart, but we observed a 50% reduction in their fertility rate in heterozygous mating of 9 months old females compared to *ap1g1*^+/+^. At the same age (9 months), in ten different matings, we observed a reduced number of laid eggs. In a representative experiment only, 52 (SD +/− 0.6) of *ap1g1*^+/−^ eggs were deposed compared to 131 (SD +/− 0.8) of *ap1g1*^+/+^. We also analyzed the percentage of fertilized eggs and observed that 91% of *ap1g1*^+/+^ eggs were fertilized compared to 52% of *ap1g1*^+/−^ (Figure 3A–C). It was reported that mice with hypomorphic mutation of the Ap1g1 adaptin gene showed dysmorphic testis with reduced numbers and motility of the sperm [10]. We thus focused on zebrafish *ap1g1*^+/−^ males and observed a 50% reduction in sperm concentration and those sperms showed a 40% reduction in motility (Figure 3D,E). The mRNA for the γ1 gene in the ovary of *ap1g1*^+/−^ fish was reduced by about 30% when compared to *ap1g1*^+/+^; in the testis of *ap1g1*^+/−^ fish, the reduction was more dramatic (50%) compared to *ap1g1*^+/+^ (Figure 3F). At 9 months mating of +/− males with +/+ females produced a female/male ratio of 2.7 whereas mating of +/− females with +/+ males produced a female/male ratio of 1 (Figure 3G). We analyzed gonads histology at 6 months of age. The +/+ females presented normal ovaries with follicles of all developmental stages, ranging from primary to full-grown stages. The +/− female ovaries revealed an impairment of follicles maturation with a reduction in eggs. They showed a disorganized ovary with degenerative oocytes and a high proportion of early and late atretic follicles and debris (Figure 3H), suggesting alteration of follicle maturation. Concerning male gonads at 6 and 9 months, *ap1g1*^+/−^ testes were smaller than control and the reduced amount of sperm emphasizes impaired reproductive capabilities of *ap1g1*^+/−^ animals of both sexes. In conjunction with the zebrafish phenotype reported here, we might assume that also uterus and/or testes function in mammals are impaired with increasing age, if AP1G1 levels are only 50% of the wild type.

### 2.4. Loss of Ap1g1subunit Affects the Development of Intestinal Epithelium in Heterozygous Fish

During adult age, as an indicator of development and growth, we measured the body length (from the head to the end of the caudal fin) and the body mass of *ap1g1*^+/+^ and *ap1g1*^+/−^ animals (males and females) at 3, 6, 9, 12 and 18 months. Concerning the body length, we did not observe any differences, but the body mass analysis enhanced differences for ap1g1^+/+^ compared to *ap1g1*^+/−^ animals, as shown in Figure 4A. In particular, histological analysis of animals at 9 months of age revealed morphological abnormalities in the intestinal epithelium. Differences can be found in the morphology of the mucosa columnar epithelial cells and the globet cells suggesting functional differentiation. Intestine mucosa of *ap1g1*^+/−^ animals displayed a sloughing epithelium at the fold tips sigh of mucosa detachment and disorganization of lamina propria. (Figure 4B). Previously, we studied zebrafish μ1-adaptins μ1A, μ1B and μ1C encoded by *ap1m1*, *ap1m2* and *ap1m3* genes, respectively, using antisense morpholino techniques. Of them, only the μ1B knock-down revealed disturbed gut development [28]. The mouse μ1B expression is limited to polarized epithelial cells and μ1B mice KO also caused defects in the gut epithelium leading to malnutrition and infections with gut bacteria. It is not known whether this phenotype is caused by the deficiency of γ1AP1/μ1B or γ2AP1/μ1B complexes or both. γ1AP1/μ1B are primarily required for endosomal basolateral protein sorting. However, μ1B expression in μ1A KO MEF cells demonstrated that the γ1AP1/μ1B complex may substitute γ1AP1/μ1A in mannose-6-phosphate receptor-mediated lysosomal protein sorting. Indeed, γ1AP1/μ1B complexes bind to the trans-Golgi network, next to γ1AP1/μ1A complexes. The μ1B heterozygous mice do not show any phenotypes and, thus, 50% of μ1B levels are sufficient to sustain gut function while the γ1AP1 heterozygous mice showed a retarded growth rate during nursing and the established body mass difference was maintained in adult animals. Based on the zebrafish data, one can assume that this phenotype is caused by delayed gut development and malnutrition in the first 6–9 months of life and, in addition, it demonstrates that a reduction γ1AP1/μ1A complexes fulfil indispensable functions in gut development because μ1B heterozygosity does not produce a phenotype. We cannot exclude that γ2AP1/μ1B-deficiency contributes to the gut phenotype of μ1B KO mice and μ1B antisense morpholino zebrafish. A γ2-adaptin knock-out zebrafish model should be generated as it will be helpful to answer this question.

### 2.5. Impaired Neuronal Development of Ap1g1^+/−^ Zebrafish Embryos

Neurons are highly polarized cells with morphologically and functionally distinct neurites, dendrites and axons which contain different proteins for specialized functions. AP-1 complex has been implicated in the polarized sorting of a subset of transmembrane receptors and transporters to the somatodendritic domain [4,32]. The previous study using zebrafish as an animal model showed that the transcript which encodes for γ1 AP-1 subunit is expressed in different areas of the developing central nervous system (CNS) from 16 to 72 hpf; in particular, ap1g1 expressions were detected in the midbrain, hindbrain telencephalon, midbrain–hindbrain boundary, but also in dorsal and ventral cells of the spinal cord, presumably Rohon–Beard sensory neurons and motoneurons [16]. To gain insight into the involvement of the γAP1 subunit during neural development, we took advantage of transgenic lines expressing EGFP under the neuronal differentiation 1 (*neurod1*) gene promoter Tg(*neurod1*:EGFP), a transcription factor known to play important roles in post-mitotic neurons during neurodevelopment of different areas as telencephalon, diencephalon, hindbrain, midbrain, midbrain–hindbrain boundary, trigeminal ganglia, olfactory placode, retinal neuroepithelial and caudally in spinal cord neurons [33,34,35]. To this purpose, we crossed the heterozygous fish *ap1g1^+/−^* (carrying the insertion of two nucleotides) with the Tg(*neurod1*:EGFP). Individuals of the F1 progeny were selected based on the contemporary presence of ap1g1 mutation and EGFP expression. The larvae were raised to adulthood to establish the stable transgenic/knock-out line, hereinafter named Tg(*neurod1*:EGFP-ap1g1). As shown in Figure 5, at 48 hpf, *neurod1*-driven fluorescence was evident in the anterior part of CNS corresponding to telencephalon (tel), midbrain (mb), hindbrain (hb), midbrain–hindbrain boundary (mhb), spinal cord neurons (scn) where cells positive for EGFP expression were detected in Tg(*neurod1*:EGFP ap1g1 ^+/+^). In general, Tg(*neurod1*;EGFP ap1g1^+/−^) embryos had a similar pattern of EGFP expression compared to Tg (*neurod1*:EGFPap1g1 ^+/+^) counterpart, but specific brain regions or cells were not decorated: fluorescence was clearly missing in the midbrain and in particular at the level of the midbrain–hindbrain boundary and also in anterior part as telencephalon. At 48 hpf in the trunk, spinal cord neurons (scn) were expressing EGFP, but we observed a marked decrease in the number of snc in heterozygous embryos compared to control embryos (+/+) (Figure 5E). The obtained results were confirmed by Light-Sheet microscopy-based 3D visualization (Video S1). Patients with mutations in the AP1G1 subunit have been reported to exhibit a small head size and microcephaly; to determine whether *ap1g1*^+/−^ fish display reduced head size, we measured body length and head size by taking anterior and posterior width measurements at 6 dpf (Figure 6A). Although we observed a mild decrease in the body length of *ap1g1*^+/−^ (13.62 mm ± 0.0310 mean ± s.e.m.) compared to *ap1g1*^+/+^ fish (11.89 mm ± 0.022 mean ± s.e.m.) (Figure 6A,B), there was a significant reduction in the width of the head in *ap1g1*^+/−^ (anterior head width, 0.0719 mm ± 0.001, and posterior head width, 0.130 mm ± 0.001 mean ± s.e.m.) compared to *ap1g1*^+/+^ fish (anterior head width, 0.0780 mm ± 0.001, and posterior head width, 0.135 mm ± 0.001; mean ± s.e.m.) at 6 dpf (Figure 6B), recapitulating microcephaly phenotypes observed in patients. We also measured brain volume in *ap1g1*^+/−^ zebrafish at 6 dpf using Light Sheet microscopy; we recorded a 35% reduction (*p* < 0.01) in the total brain volume of *ap1g1*^+/−^ compared to *ap1g1*^+/+^. The reconstruction of the 3D distribution of EGFP fluorescence in 48 hpf +/+ and +/− embryos clearly evidenced the lack of EGFP signal in the anterior part of the CNS, in the hindbrain region and in particular in the midbrain–hindbrain boundary (Figure 6C–G). Altogether, the analysis of the Tg(*neurod1*:EGFPap1g1^+/−^) line suggests that normal expression of *ap1g1* at early stages of neurodevelopment is required for determination and/or differentiation of specific neuronal populations located in different brain areas and spinal cord neurons.

### 2.6. Reduced E-Cadherin and BMPs Expression in Zebrafish Ap1g1^+/−^ Larvae

Critical steps in early vertebrate development are the morula to blastula transition and the development of the blastocyst are critically dependent on the adherens junction (AJ) protein E-cadherin. Gastrulating cells may use E-cadherin-mediated cell–cell contact as a migration substrate and real movements that drive convergence and extension. In addition, cadherin molecules are necessary to maintain the integrity and survival of precordal plate derivatives and cellular adhesion between epithelial cells; failure to maintain these structures may have resulted in axis pattern defects due to reduced organized activity of different pathways. The morphological alterations of the *ap1g1*^−/−^ zebrafish blastula, the inability of *Ap1g1*^−/−^ mouse blastocysts to hatch out of the zona pellucida and the inability of their ES-cells to adhere to feeder cells and to grow in cell culture strongly suggested AJ dysfunction. Morula of E-cadherin KO mice undergo compaction, but the cell–cell contacts are not stable and single cells can be lost, but the development can proceed to the blastocyst stage. However, these blastocysts are unable to hatch out of the zona pellucida, just like the *Ap1g1*^−/−^ mouse blastocyst [8,36]. Real-time qPCR analysis evidenced a significant reduction of E-cadherin mRNA level in *ap1g1*^+/−^ embryos at 48 hpf compared to controls. This indicates that the reduced fertility of zebrafish *ap1g1*^+/−^ females is not only due to the reduced production of eggs, but also due to the production of eggs with much-reduced mRNA and protein contents, which may not suffice to perform the earliest developmental stages and formation of epithelial polarized cells. E-cadherin +/− KO mice do not show a phenotype in blastocyst development and thus the lower E-cadherin level in *ap1g1*^+/−^ zebrafish should suffice to mediate blastula development. Thus, failure of E-cadherin sorting to and maintenance of AJ, impaired by the lack of *ap1g1* mediated E-cadherin sorting, appears to be responsible for the failure of blastocyst development. The resulting E-cadherin miss-sorting may lead to its trapping in a TGN-endosomal compartment, contributing to or causing the increase in protein over mRNA level. The BMP signalling cascade plays a central role in dorso-ventral patterning of zebrafish embryos. We know that Bone Morphogenetic Proteins (BMPs) are extracellular proteins which belong to Tgfβ superfamily. During zebrafish gastrulation, different bmp genes are expressed: bmp2a, bmp4 and bmp7 that are essential for embryonic development, organogenesis, tissue specification and regeneration. BMP4, in particular, has a variety of biological functions and plays a prominent role in the regulation of epithelial cell differentiation [37]. As shown in Figure 7, the mRNA level of the analyzed BMP factors showed a reduction in *ap1g1*^+/−^ mutants compared to wild-type larvae.

## 3. Discussion

Cellular trafficking via transport vesicles and tubules is essential to maintain critical biological functions and thus is mediated by tightly regulated machinery. AP family complexes are components of membrane protein coats that mediate cargo selection and coated vesicle formation [10]. Vertebrates have five AP complexes and tissue-specific AP isoforms belong to the largest AP complex family. The absence of the ubiquitous AP1 and AP2 complexes is incompatible with vertebrate development, while the absence of AP3, AP4 or AP5 complexes leads to the development of adult organisms displaying specific disease phenotypes. The AP1 complexes form the largest family compared with AP2 and AP3 complexes. Isoforms of AP4 and AP5 complexes are not known. AP1 complexes have functions in protein sorting and transport at the *trans*-Golgi network and on endosomes. Due to their regulation of intracellular protein sorting, they are also indirectly involved in the regulation of protein levels in the plasma membrane [38]. The two ubiquitous AP1G1 and AP1G2 complexes share the same additional three subunits and each one fulfils specific, indispensable functions in vertebrate development. They share the additional ubiquitous and tissue-specific subunits. Most of the work on AP-1 complex function has been done using knockout and knockdown methods, but recent data from animal models and human patients demonstrate that point mutations in even ubiquitous AP1 subunits cause different diseases called adaptinopathies. Mice homozygous for a null mutation of AP1 γ1 subunit die at day 3.5 post-fertilization, while heterozygous mice display a reduced growth rate exclusively during nursing and impaired T-cell development [8]. Later, Johnson et al. [9] described the phenotype of adult mice that are homozygous for a hypomorphic mutation of Ap1g1. The new recessive mutation arose spontaneously in a colony of mice at The Jackson Laboratory, named “figure eight” (*fgt*) to reflect the atypical movement pattern (circling behaviour) of mutant mice (*fgt/fgt*). These mutant mice are viable and exhibit multiple phenotypic abnormalities, such as hearing loss and eyes, thyroid and testes defects. In humans, impaired functions of different subunits of AP complexes have been associated with severe inherited disorders. Individuals of families with variants in the *AP1G1* gene show clinical phenotypes common to neurodevelopmental disorders (NDD), including global developmental delay (GDD) and intellectual disability (ID), which varies from mild to severe. In particular, they show clinical features such as epilepsy, developmental delay, speech delay, hyperactivity and aggressive behaviour. Additionally, recessive loss-of-function variants in *AP1B1* cause a multi-organ disorder with clinical manifestations such as enteropathy, hearing impairment, peripheral neuropathy, keratodermia encephalopathy and intellectual disability [23]. *AP1S1* mutations cause MEDNIK (mental retardation, enteropathy, deafness, peripheral neuropathy, ichthyosis, and keratoderma) syndrome [21]; AP1S2 mutations are responsible for some forms of X-linked mental retardation [18]; and AP1S3 mutations increase susceptibility to pustular psoriasis [22,39]. These recent data make it necessary to study AP1 adaptin functions in more detail using model organisms. The zebrafish model, despite being a non-mammalian system, shows high homology at the genomic level and high similarity in brain development and structures; the presence of similar functions and high similarities with other tissues makes it a suitable model for preclinical studies [40]. We have demonstrated that zebrafish is a suitable model system to study AP1 functions in vertebrates. Here we generated a zebrafish knockout model for the AP1G1, the γ1 subunit, using the CRISPR/Cas 9 technique. We introduced an insertion of 2 nucleotides, which causes a frame shift in the open reading frame and hence a truncated protein of only 26 amino acids. The mutation was inherited with the Mendelian pattern and when we genotyped the F2 generation, we observed that the *ap1g1^−/−^* embryos ceased their development at 3.5 hpf during the transition period of the blastula-gastrula stage. On the contrary, the heterozygous fish reached the adult stage, but they showed a reduced growth rate and reductions in their fertility rate. Males and females have defects in spermatogenesis and oocyte maturation, respectively. One-year-old males showed a reduction in sperm concentration and motility, while females showed impairment in follicle maturation. The zebrafish *ap1g1* knock-out model revealed the indispensable function of the family of AP1 complexes for vertebrate development and that heterozygosity causes disease phenotypes. This result confirms the suitability of the zebrafish model system because these data are in line with the phenotypes observed in the *Ap1g1* knock-out mouse model. The *Ap1g1^−/−^* mice ceased development prior to hatching out of the zona pellucida and when these blastocysts were seeded onto feeder cells, they did not adhere to the feeder cells and did not proliferate. The molecular basis of this phenotype was not analyzed at the time. The *Ap1g1*^+/−^ inner cell mass cells grew just like isogenic wild-type cells [8]. The fertility of *Ap1g1*^+/−^ mice was not quantified, but they reproduced less efficiently than the isogenic wild-type mice. However, the *Ap1g1*^+/−^ mice grew slower, specifically during nursing than the wt control animals. In addition, they had a defect in the formation of CD4 single-positive T-cells, while CD8 single positive T-cell numbers were normal. These data point to impaired precursor cell differentiation processes. In zebrafish, we observed a strong reduction of E-Cadherin and BMP factors. Morula of E-cadherin knockout mice do undergo compaction, but the cell–cell contacts are not stable and single cells are lost. The development does proceed to the blastocyst stage, but these blastocysts are unable to hatch out of the zona pellucida [36]. The zebrafish knockout, similarly, to the *Ap1g1^−/−^* KO mouse blastocysts, also has this phenotype. E-cadherin has functions in numerous tissues. Tissue-specific functions have been observed in almost every epithelium. E-cadherin sorting plays a role in the establishment and maintenance of adherens junction (AJ) complexes. In E-cadherin heterozygous animals, only very few tissues show morphological alterations. These are the gut, ovaries and brain. In *Drosophila melanogaster*, cell-type specific AP1 γ1 knockdowns, E-cadherin is transported to cell–cell contact sites, but E-cadherin is not able to support cell–cell contacts upon increasing plasma membrane tension [41]. This indicates insufficient E-cadherin targeting the plasma membrane and adherent junction. E-cadherin, N-cadherin and VE-cadherin (cadherin-1, -2 and -5, respectively) have highly conserved cytoplasmic domains with conserved putative canonical sorting motif sequences for AP1-mediated sorting and transport. A cytoplasmic di-leucine sequence important for E-cadherin sorting is located directly after the trans-membrane domain [42,43,44]. The impaired uterine and gonadic functions of *ap1g1*^+/−^ fish may also be caused by defects in the organization of AJ due to inefficient cadherin sorting [45,46]. The development of *ap1g1*^+/−^ heterozygous embryos till adult age, indicates that ap1g1 expression levels suffice to permit cadherin-mediated AJ functions. However, cadherins have functions in signal transduction pathways, which are independent of their functions in AJ and one can speculate that those activities may not suffice for proper tissue development [47]. AJ are metastable and are continuously disassembled and reorganized. Therefore, two major cadherin transport routes contribute to AJ biogenesis and restructuring [41,48]. Firstly, cadherins are transported from the biosynthetic pathway and the TGN to the plasma membrane. Secondly, cadherins are endocytosed and recycled via endosomes back to plasma membrane AJ. Given the prominent AP1G1 function in sorting proteins at the TGN, one could expect that the primary AP1G1-dependent cadherin sorting and transport step takes place at the TGN. All data support AP1G1-dependent sorting and transport of cadherins from the TGN to the plasma membrane, because AP1G1 sorting of cadherin is most important during the biogenesis of tissues [41,48]. In the absence of AP1G1, constitutive, randomized and possibly slow cadherin export to the plasma membrane domains appears to be sufficient for the formation and maintenance of a limited number of AJ. After arriving at the plasma membrane, cadherins could be sorted independently of AP1G1 via endosomal recycling to dynamic AJ. Under conditions of most active cadherin plasma membrane-endosome recycling, due to its higher steady-state concentration in early endosomes, a larger cadherin fraction may be diverted from early endosomes into endolysosomes for degradation. These cadherins will have to be replaced by cadherins from the biosynthetic pathway. Thus, impaired sorting in both pathways enhances the importance of the biosynthetic pathway for AJ formation and function. It has been proposed that the E-cadherin endosomal recycling pathway is mediated specifically by the AP1G1/AP1M2 (γ1AP1μ1B) complex, which is only expressed in polarized epithelial cells [10,49]. However, the limitation of the AJ phenotypes to highly proliferative tissues with highly active cell–cell contact biogenesis indicates a more critical function of AP1G1 in E-cadherin sorting in the biosynthetic pathway compared to a recycling pathway. The AP1G2 complex, which contains the same cargo binding μ1A, μ1B and σ1A and σ1B adaptins, is not able to substitute AP1G1 in neurons, E-cadherin sorting and transport. The heterozygous fish showed a reduced body mass at different ages in a range between 3 and 18 months post-fertilization. Additionally, histological analysis showed severe abnormalities in the intestinal mucosa. The barrier and transport functions provided by epithelia depend on their highly-polarized phenotype. Polarization of epithelial cells is a complex process directed by external cues such as cell–cell and cell–extracellular matrix (ECM) interactions. These interactions initiate the formation of specialized junctions that demarcate apical and basolateral plasma membrane domains. Once polarity is established, a different array of cellular machinery is used to ensure the maintenance of epithelial polarity, with specialized intracellular trafficking pathways directing the domain-specific targeting of newly synthesized, endocytosed and transcytosed molecules. The accurate delivery of molecules to their appropriate membrane domains depends on the differentiation state of the cell. Consequently, defects in trafficking pathways that are used to maintain epithelial polarity or alterations in epithelial differentiation can cause disease in organs in which epithelial cell polarity is crucial, such as the intestine, liver and kidney. In the brain of all vertebrate models, the vesicular protein transport pathways are highly conserved. The brain is the tissue with the most active vesicular protein transport. The AP1 complex is proven to be essential for synaptic vesicle protein (apical) recycling and the polarized sorting of somatodendritic proteins with the soma or retrograde retrieval from axons. Interference with AP1 complex function results in a reduced number of synapses and morphological defects in dendritic spines [50,51]. In our knockout model, the heterozygous fish displayed abnormalities in specific brain areas and spinal cord neurons, and the brain volume is reduced. This phenotype confirms the contributions of the AP1 complex in polarized epithelial cells and neurons; a more detailed investigation of E-cadherin sorting steps would contribute to understanding this phenotype. The neuronal phenotype was evident in *ap1g1* heterozygous animals and is associated with markedly reduced growth during nursing and reduced fertility. In conclusion, our data demonstrate that zebrafish is a suitable model system to study the molecular mechanisms causing human AP1-mediated adaptinopathies. The study highlighted the important role of the subunit in different organs and tissues and its function in polarized cells as neurons or epithelial tissues. The findings might help to understand why the adaptinopathies have a common clinical phenotype. However, it also might suggest deeper studies in patients in order to clarify potential defects in polarized cells. The study of the biological basis of these kinds of diseases could provide relevant topics to better characterize the pathogenesis and treat adaptinopaties.

## 4. Materials and Methods

### 4.1. Maintenance and Handling of Zebrafish Lines

All experiments were performed in accordance with the Italian and European Legislations (Directive 2010/63/EU)9 and with permission for animal experimentation from the Local Committee for Animal Health (Organismo per il Benessere Animale) of the University of Brescia and the Italian Ministry of Health (Authorization Number 287/2018-PR). *Danio rerio* (zebrafish) were maintained in 3 L of water at a temperature-controlled (28.5 °C) with 14 h light-and 10 h dark cycle and fed as described by Kimmel and colleagues (Zebrafish: Housing and husbandry recommendations [52,53]. Mating was set up in organizing tanks, upon fertilization the eggs were collected and placed in a Petri dish containing fish water and incubated at 28 °C. For anaesthesia of zebrafish embryos and larvae, tricaine (MS222; E10521, Sigma–Aldrich St. Louis, MO, USA) was added to the fish water at 0.16 mg/mL. The wild-type AB line used in this work included an AB strain (KIT Institute -Karlsruhe-Germany) and a transgenic line Tg(*neurod1*-EGFP) [54].

### 4.2. Generation and Genotyping of Zebrafish ap1γ1 Mutant Line

We used the CRISPR/Cas9-mediated genome editing technique to generate the *ap1g1* mutant lines following the protocol of Gagnon et al., 2014. Single guide RNA (sgRNA) was designed using the CHOPCHOP algorithm (https://chopchop.cbu.uib.no, accessed on 1 March 2015), to specifically target an optimal CRISPR sequence on exon 1 of γ1ap1 gene (NM_199682.1). The chosen sgRNA (see Appendix A) was transcribed in vitro using the MEGAshortScript T7 kit (AM1354, ThermoFisher Scientific, Waltham, MA, USA) from a double-strand oligonucleotide template. One-cell stage embryos were injected with 2 nL of a solution containing 300 ng/μL of Cas9 protein (M0646, New England Biolabs, Ipswich, MA, USA) and 150 ng/μL of sgRNA; phenol red was used as an injection marker. Genomic DNA was extracted from single 5 dpf larvae to verify the presence of mutation and confirm the activity of the sgRNA. F0 injected embryos were raised to adulthood and founders screened, then crossed with AB strain to generate F1 offspring and to confirm the germline transmission of the mutation. Heterozygous mutants, harbouring the mutation of choice, were outcrossed 2 times and then incrossed to obtain homozygous mutants (F3 generation). Screening primers for heterozygous fish (Appendix A) were designed to amplify a 360-bp region across the ap1g1 sgRNA target region. Larvae or adult fish were anaesthetized with tricaine, and a small fragment of the caudal fin was cut with a sharp blade. Genomic DNA was extracted using the HOTShot protocol [55]. Mutations in F0 were detected using heteroduplex mobility assay (HRMA) [56]. In this case, the genomic DNA fragment of the target site was amplified by PCR using the locus-specific primers listed in Appendix A (Ap1g1FOR and Ap1g1REV) that gave rise to a 360 bp fragment. PCR Conditions were as follows (94 °C 5 min; 94 °C 30 s; 60 °C 30 s; 72 °C 30 s) (37 cycles). The resulting PCR amplicons were electrophoresed on 15% non-reducent polyacrylamide gel. For verifications, PCR products from fish harbouring indel mutations were subjected to sequencing. Poly Peak Parser software 2014 (http://yost.genetics.utah.edu/software.php, accessed on 2 April 2023) was used for the identification and sequence characterization of heterozygous mutants generated by the CRISPR/Cas 9 technique.

### 4.3. Generation of Heterozygous ap1g1 Mutant Transgenic Zebrafish Line

The *ap1g1^+/−^* line was crossed with the Tg(*neurod1*-EGFP) transgenic line [54]. The resulting transgenic mutant line was analyzed to study the localization of the EGFP signal. Fluorescence analysis was performed on sibling larvae and was followed by genotyping.

### 4.4. Mendelian Analysis

At 3 months, we crossed *ap1g1^+/−^* individuals and genotyped (as described above) at different developmental stages: 3.5 hpf, 3 dpf, 28 dpf and 90 dpf.

### 4.5. Whole-Mount In Situ Hybridization (WISH)

*ap1g1^+/+^*, *ap1g1^+/−^* and *ap1g1^−/−^* zebrafish embryos were collected at 3 hpf and fixed with 4% (*v*/*v*) paraformaldehyde (PFA) (Sigma-Aldrich, St. Louis, MO, USA), dehydrated in 100% (*v*/*v*) methanol, and stored at −20 °C. The WISH was performed as previously described in [33] and selected embryos after treatment with proteinase K (10 µg/mL Merk KGaA, Darmstadt, Germany) were hybridized overnight at 68 °C with the desmoplakin b probe (*dspb*) DIG-labelled. Embryos were washed with ascending scale of Hybe Wash/PBS and SSC/PBS and then incubated with anti-DIG antibody conjugated with alkaline phosphatase overnight at 4 °C. The staining was performed with NBT/BCIP (Blue staining solution, Merk KGaA, Darmstadt Germany) alkaline phosphatase substrates. Images were taken with Zeiss Axio Zoom V16 equipped with Zeiss Axiocam 506 colour digital camera and processed using Zen 3.5 (Blue Version) software from Zeiss (Oberkochen, Germany) The experiments were done two times with *n* = 50 for each genotyped condition.

### 4.6. Histological Analysis

Wild type and *ap1g1^+/−^* mutants were fixed for 24 h in Bouin’s solution (picric acid, acetic acid and formaldehyde) at room temperature. Samples were dehydrated through graded series of ethanol, infiltrated with xylene and embedded in Paraplast plus (39602004, Leica). The samples were serially cut into 7–8-μm sections on an LKB microtome. After rehydration, the sections were stained with hematoxylin and eosin for microscopic analysis. The images were acquired using an optical microscope (Olympus, Hamburg, Germany).

### 4.7. RNA Extraction and qPCR

Total mRNA was extracted from 30 larvae at different developmental stages using TRI-Reagent (Merk KGaA, Darmstadt, Germany) according to the manufacturer’s protocol. RNA was quantified using My Spect spectrophotometer (WWR International, Radnor, PA, USA) and controlled by electrophoretic separation on a 1% TAE-agarose gel. 1.5 μg of total RNA was retrotranscribed to cDNA using Im-Prom Reverse Transcriptase (Promega, Madison, WT, USA) and oligo (dT) primers. Primers were designed by the real-time PCR tool from IDT and the sequences are reported in Appendix A. Real-time PCR was performed in triplicate using the Applied Biosystems ViiA 7 Real-Time PCR System. Reactions were performed in a 10 μL volume, with 0.5 μM of each primer, 5 μL of Syber Green Master Mix (Biorad, Hercules, CA, USA) and 75 ng of cDNA. The amplification profile consisted of a denaturation step (95 °C for 1 min) followed by 40 cycles at 95 °C for 15 s, 60 °C for 30 s) followed by a melting cycle. Each reaction was performed in triplicate, and the relative expression of each gene was calculated with the ∆∆Ct method, using *rpl13a* as a reference gene.

### 4.8. Ejaculate Collection

Fish were isolated for 2 days before stripping, to guarantee the replenishment of sperm reserve. Males were anaesthetized in Tricaine 1X (0.16 mg/mL), gently dried on a paper towel and placed in a dampened sponge ventral side up, with their genital papilla exposed. The ventral surface of the fish was further dried to remove any excess water that could prematurely activate the sperm. The fish abdomen was gently pushed with soft plastic tweezers and the ejaculate was collected from the genital papilla by means of a Drummond micro dispenser equipped with a 5 μL microcapillary tube. The whole ejaculate was diluted and preserved until analyses (within 1 h) in 20 μL of zebrafish sperm immobilizing solution (ZSI: 140 mM NaCl, 10 mM KCl, 2 mM CaCl_2_, 20 mM HEPES, pH = 8.5) 21. A total number of 8 males, 4 with an *ap1g1^+/+^* genotype and 4 to *ap1g1^+/−^* were stripped.

### 4.9. Sperm Concentration

Sperm counts were performed using an improved Neubauer hemocytometer under 400× magnification, after properly diluting 2 μL of a subsample taken from the whole ejaculate maintained in ZSI. The sample was gently mixed with a micropipette before filling the chamber. The average of five counts per sample was used to estimate sperm concentration, considering dilution steps, and expressed as a number of spermatozoa/mL.

### 4.10. Sperm Velocity

Sperm were activated by gently mixing with a Gilson micropipette 1 μL of ejaculate preserved in ZSI with 10 μL of aged tap water. Three microliters of samples were then quickly placed in separate wells on a 12-well multi-test slide and covered with an 18 × 18 mm coverslip. Both slide and coverslip were previously coated with 1% polyvinyl alcohol (Sigma–Aldrich), to avoid sperm sticking to the glass slide Sperm velocity was measured using a CEROS Sperm Tracker (Hamilton Thorne Research, Beverly, MA, USA) in the first 10 s after activation. The measurement was repeated on three subsamples per male, leading to a repeatability of 0.68 ± 0.1422, and mean values were used for statistical analyses. Mean velocity measurements were based on 280 ± 169 (mean ± s.d.) sperm tracks per male. Since sperm velocity parameters (VAP: average path velocity, VSL: straight-line velocity, and VCL: curvilinear velocity) were highly correlated (all Pearson R > 0.88; all *p* < 0.001), only VAP was considered in the following analyses. The measurement was repeated twice per male, yielding a repeatability of 0.68 ± 0.14, and mean values were used for statistical analyses.

### 4.11. Acridine Orange Staining

To analyze the level of cell death, acridine orange staining was performed using a standard protocol [57]. Embryos at 48 hpf were dechorionated and incubated for 30 min in acridine orange solution (10 mg/mL). Embryos were rinsed three times using PBS, mounted using 80% glycerol, quickly imaged using epifluorescent microscopy (Zeiss Axio Zoom V16 equipped with Zeiss Axiocam 506 colour digital camera and processed using Zen 3.5 (Blue Version) software from Zeiss (Oberkochen, Germany).

### 4.12. Microscopy

Bright-field imaging of embryos and larvae (anaesthetized with tricaine 0.16 mg/mL embedded in 0.8% low melting agarose and mounted on a depression slide) were captured using a Zeiss Axio Zoom V16 equipped with Zeiss Axiocam 506 colour digital camera and processed using Zen 3.5 (Blue Version) software from Zeiss (Oberkochen, Germany) Confocal Images were acquired using a Plan-Neofluar 10X/0.3NA objective. A Z-stack was performed to acquire the whole central nervous system and Maximum Intensity Projection was obtained using Zen Black software. For Light Sheet analysis embryos were first anaesthetized using tricaine (0.02% in fish water) and subsequently included using in a low melting agarose matrix (Top Vision Low Melting Point Agarose, Thermo Fisher Scientific) at 0.5% in fish water. Images were acquired using Zeiss LightSheet microscope V1 supported by ZenPro software using a 488–30 nm laser and 505–545 nm filter. Images from the same experiment were taken with the same laser intensity and exposure time to generate comparable images. After the acquisition, 3D images were generated and manipulated using Arivis Vision 4D (Zeiss Oberkochen, Germany) 3D reconstructions of EGFP-positive cells were manipulated to obtain pictures comparable to each other in terms of fluorescence intensity. 3D reconstructions were exported as a single snap with the same compression setting. Voxel-wise comparisons were performed following the protocol of [58] only on pixels within a mask encompassing the brain, excluding non-neural tissue and background. 

### 4.13. Statistical Analysis

All the experiments described in the manuscript were performed at least two or three times using GraphPad Prism V8 (Dotmatics, Boston, MA, USA) to perform statistical analysis. The comparison and significance between different groups were determined by one-way ANOVA, corrected for multiple comparisons or by two-tailed unpaired Student’s *t*-test. The *p*-value is indicated with asterisks * *p* < 0.05, ** *p* < 0.01, *** *p* < 0.001, and **** *p* < 0.0001. Differences were considered significant at *p*-values of less than 0.05.

## Figures and Tables

**Figure 1 ijms-24-07108-f001:**
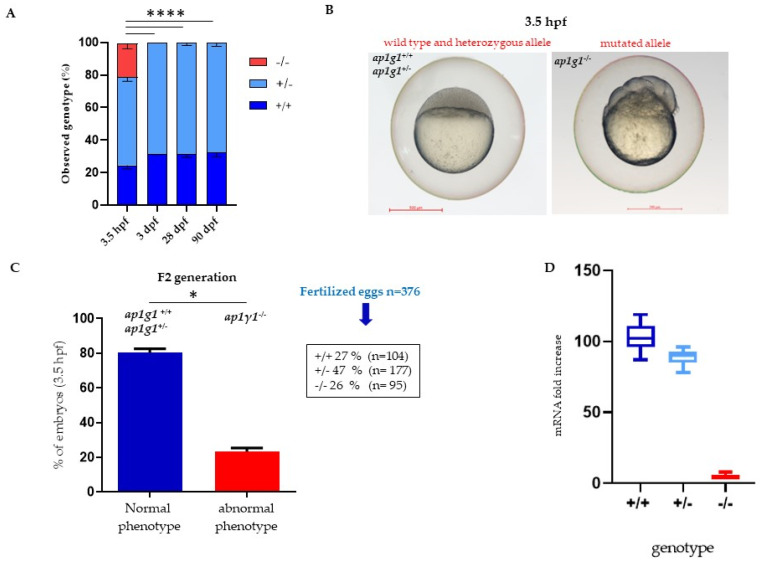
Characterization of *ap1g1* mutant embryos and larvae. (**A**) Representative images of genotype distribution at different developmental stages. Data derived from five different matings with *n* = 100 in each mating. For genotyping see Math and Met Section. (**B**) Comparison of the morphological phenotype of embryos derived from F2 generation (F1 *ap1g1*^+/−^ × F1 *ap1g1*^+/−^) at 3.5 hpf. (**C**) In the graph blue bar indicates the percentage (%) of embryos with +/+ and +/− genotype and normal phenotype, the red bar indicates the embryos with −/− genotype and abnormal phenotype. The data represent one experiment out of three with *n* = 376 embryos. (* *p* < 0.05, **** *p* < 0.0001 One-way ANOVA + Newman-Keuls). Size bar = 500 μm; (**D**) qPCR analysis of *ap1g1* transcript from *ap1g1*^+/+^; *ap1g1*^+/−^ and *ap1g1*^−/−^ embryos at 24 hpf. Gene expression was normalized using *rpl13a* as a reference gene and expressed as the mRNA fold increased.

**Figure 2 ijms-24-07108-f002:**
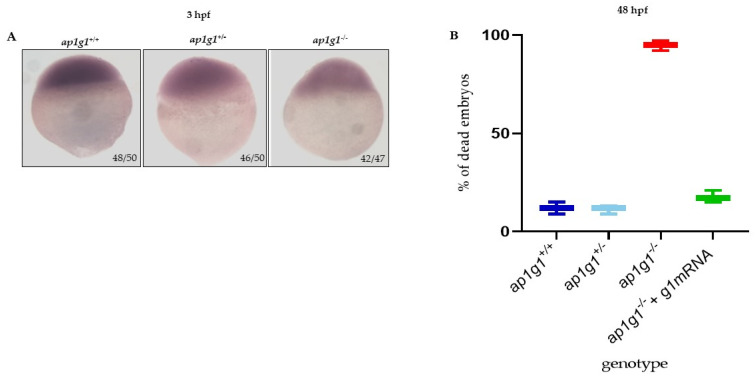
Characterization of ap1g1 mutant embryos. (**A**) Whole mount in situ hybridization (WISH) for *dspb* performed in embryos at 3.5 hpf. The experiment was done three times with *n* = 25 embryos at each experiment. Ratios at the bottom-right part of each picture specify the number of embryos showing the same staining pattern, compared to the total number of embryos used for each experiment. The images were taken in dorsal position at 32× magnification with a Zeiss Axiozoom V13 (Zeiss, Jena, Germany) microscope, equipped with a PlanNeoFluar Z1×/0.25 FWD 56 mm lens and Zen Pro software. (**B**) The graph in the panel shows, at 48 hpf, the percentage of dead embryos of the three different genotypes and the percentage of dead embryos after the microinjection with zebrafish wild type γ1 mRNA. The experiment was repeated three times.

**Figure 3 ijms-24-07108-f003:**
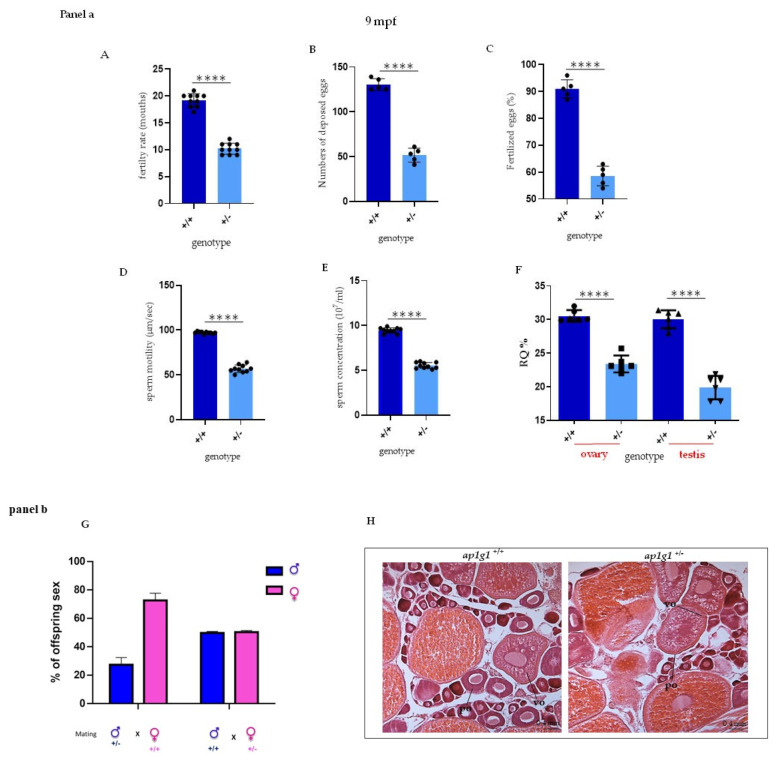
Fertility analysis in heterozygous *ap1g1^+/−^* fish. Panel a (**A**) The graph represents the fertility rate of heterozygous compared to +/+ females. The data are representative of five different experiments with 10 fish per group (**B**,**C**) The graphs show respectively the number of laid eggs and fertilized eggs in +/+ and +/− females at 9-mouths-old fish obtained by three different experiments; the observed female was 12 for each group. Concentration (**D**), velocity (**E**) and viability (**F**) of ejaculated sperm. Depicted is mean ± SEM. **** *p* < 0.0001; *n* = 10, in triplicate. Motility (**D**) and concentration (**E**) of ejaculated sperm from 1-year-old fish. Experiment was done twice. Motility: *n* = 10 +/+ mean = 97.20 and *n* = 10 +/− mean = 56.80; concentration: *n* = 7 +/+ mean = 9.46 × 10^7^/mL and *n* = 7 +/− mean = 5.54 × 10^7^/mL. (**F**) qPCR analysis on +/+ and +/− fish respectively in ovary and testis. The experiment was repeated twice. *n* = 10 for each column. Panel b (**G**) The graph represents the percentage of different offspring obtained by matings between *ap1g1*^+/−^ male and a*p1g1*^+/+^ female and ap1g1^+/+^ male with ap1g1^+/−^ female. (**H**) Histological analysis of *ap1g*1^+/+^ and *ap1g1*^+/−^ ovary detects oocyte maturation defects in mutant females; black arrowheads indicate debris of atretic follicles. Abbreviations: po primary oocyte, vo vitellogenic oocyte.

**Figure 4 ijms-24-07108-f004:**
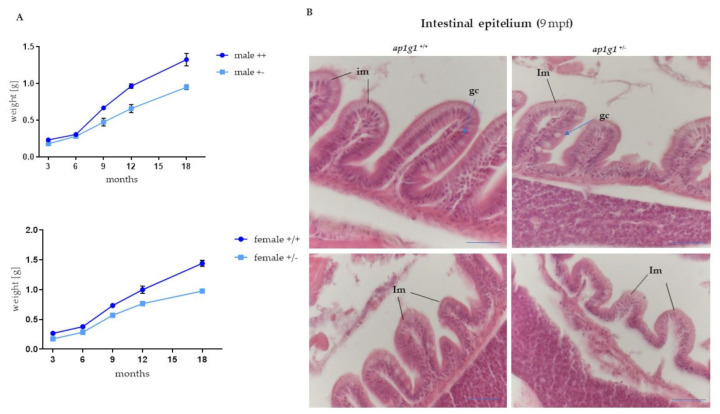
Histological analysis of intestinal epithelium in adult fish. (**A**) Growth curve of ap1g1^+/+^ and ap1g1^+/−^. Body weight (in gr) was measured at different ages of adult life from 3 to 18 months. Both males’ and females’ *ap1g1*^+/−^ body weight is significantly lower compared to *ap1g1*^+/+^. Statistical significance was determined by multiple t-test comparing each age group. For each group, 10 individuals were used and data was derived from two different experimental groups. (**B**) Histological analysis of *ap1g1*^+/+^ and *ap1g1*^+/−^ intestinal mucosa with abnormal epithelium at the villous tips and reduced height of villi in the heterozygous mutant at 9 months age-old compared to wild-type homozygous fish. Scale bar = 100 μm Abbreviations: im: intestinal mucosa; gc: globet cells.

**Figure 5 ijms-24-07108-f005:**
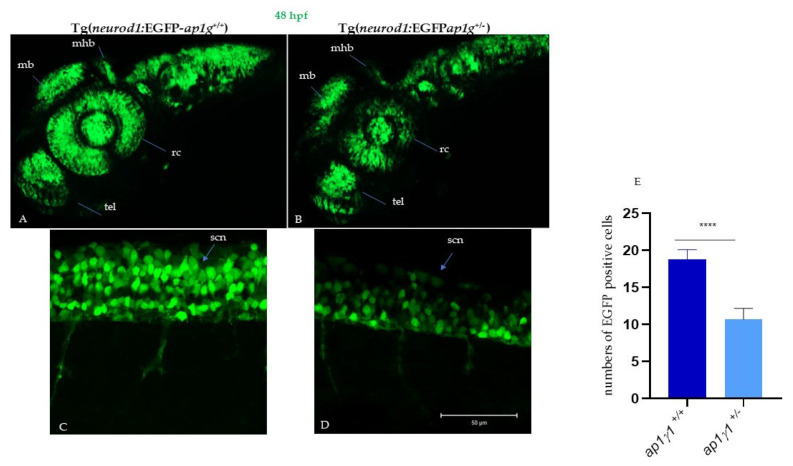
Neuronal characterization of *ap1g1^+/+^* and *ap1g1^+/−^* embryos: Confocal images of *ap1g1*^+/+^ and *ap1g1*^+/−^ embryos in transgenic background tg(*neurod1*:EGFP) 48 hpf. Lateral view of the central nervous system (**A**) and of spinal cord neurons (**B**) acquired using an LSM510 Meta confocal microscope (Zeiss (Oberkochen, Germany). (**A**,**B**) Images were acquired using a Plan-Neofluar 10X/0.3NA objective. A Z-stack was performed to acquire the whole central nervous system and Maximum Intensity Projection was obtained using Zen Black software. Abbreviations: mb, midbrain; hb, hindbrain; mhb, midbrain–hindbrain boundary; tel, telencephalon; rc, retinal cells; snc, spinal cord neurons. Single plane images were acquired using a PlanApochromat-63X/1.4 NA oil DIC objective. Size bar: 50 µm. (**C**,**D**) EGFP-positive cells were quantified by drawing polygonal regions on the image acquired and manually counting the number of cell somas in the selected area. The analysis is the result of 3 different experiments with *n* = 15 for each genotype. (**E**) Quantification of EGFP positive cells in the described spinal cord area described in figure C and D; **** *p* < 0.0001, Student’s *t*-test.

**Figure 6 ijms-24-07108-f006:**
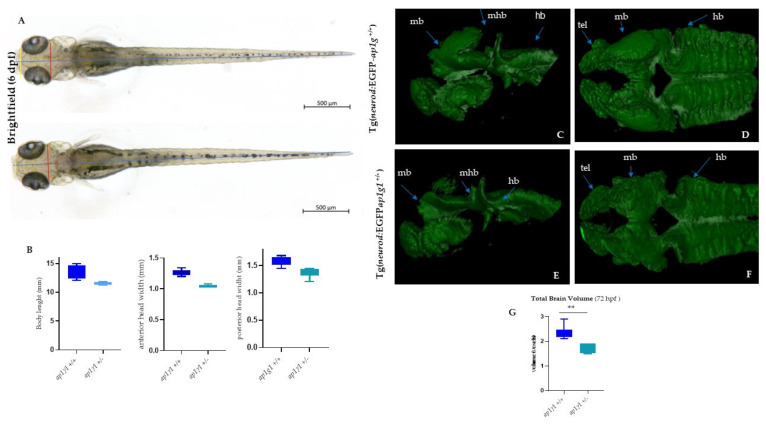
Analysis of body length, head size and 3D LightSheet analysis: (**A**) Bright-field images of *ap1g1*^+/+^ and *ap1g1*^+/−^ larvae at 6 dpf. Lines indicate body length (blue), anterior head width (yellow) and posterior head width (red). Scale bars: 500 mm (**B**) Quantification of body length, anterior head width and posterior head width measurements of *ap1g1*^+/+^ and *ap1g1*^+/−^ larvae. Data are mean from three independent experiments (*n* = 15, 13 and 16 for *ap1g1^+/+^* and *n* = 12, 12 and 9 for *ap1g1*^+/−^. 3D Light Sheet analysis at 72 hpf: Lateral view and (**C**,**E**) dorsal view (**D**,**F**) of the anterior part of the central nervous system of wild type and heterozygous embryos. Magnification 25×. (**G**), Quantification of total brain volume from. *ap1g1*^+/+^ and *ap1g1*^+/−^ embryos at 72 hpf. Data are mean ± s.e.m. for three independent experiments *n* = 15, 15 and 16 for *ap1g1*^+/+^, and *n* = 10, 10 and 10 for *ap1g1*^+/−^ embryos, ** *p* < 0.01 (two-tailed *t*-test). 1 voxel = 8 µm^3^ Abbreviations: mb, midbrain; hb, hindbrain; mhb, midbrain hindbrain boundary; rc, retinal cells; snc, spinal cord neurons.

**Figure 7 ijms-24-07108-f007:**
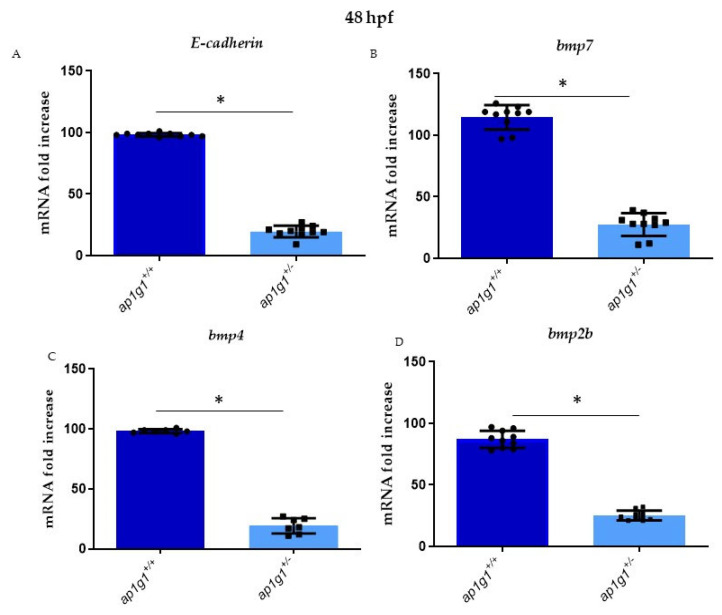
(**A**–**D**) Alterations of *E-cadherin* expression and different BMP factors in heterozygous and wild-type embryos, respectively: *E-cadherin, bmp7, bmp4 and bmp2b.* qPCR experiments were performed at 48 hpf in triplicate on cDNA samples derived from *n* = 30 whole embryos per condition to measure the mRNA level of E-cadherin and BMPs factor in *ap1g1*^+/+^ and *ap1g1*^+/−^. Gene expression was normalized using *rpl13a* as a reference gene and expressed as the mRNA fold increased. Data are representative of three replicates and are shown as the mean ± standard deviation; * *p* < 0.01 Student’s *t*-test, Statistical analysis was performed with GraphPad Version 8.3.3 GraphPad Software, Inc., Version La Jolla, CA, USA.

## Data Availability

Data are available from the corresponding author upon request.

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
