# Peer review of "Deficiency of AP1 Complex Ap1g1 in Zebrafish Model Led to Perturbation of Neurodevelopment, Female and Male Fertility; New Insight to Understand Adaptinopathies"

_ijms, 2023, doi:10.3390/ijms24087108_

Round 1

Reviewer 1 Report

The manuscript "The zebrafish ap1g1 is indispensable for fertility and neurodevelopment: a new model to understand adaptinopathies" addresses an important issue: the understanding of the functional role of AP1G1 in the mechanisms underlying the pathogenesis of a new class of neurocutaneous and neurometabolic disorders affecting intracellular vesicular traffic, known as adaptinopathies, exploiting the advantages of the zebrafish model and the CRISPR/Cas9 genome editing technology.

The objectives were clearly stated and explained in the manuscript, however the experimental strategy raises some major concerns and so the experimental information from which the conclusions were drawn. The manuscript is overall well written and has good organization. Quantitative analysis of the experimental data is missing throughout the manuscript and the interpretations of the results and the discussion are thus suffering from these limitations.

The paper is interesting but there is a need for more experimental detail in order to critically review the data. Specifically, they should provide information for the following questions and comments:

Major points:

1.      The authors should include more recent update on this topic and compare how this study further advances the current knowledge in the “Introduction section”.

2.      Resolution of the bar graphics may be improved for the shake of clarity. Specially in Figures 2,6 and 7. And they should also be resized if appropriate.

3.      The Methods section in the study should be more accurately described for each technique used.

Minor points:

1.      Unify the style of the references in the References Section and add DOI in the cases it is possible. And use the same reference and citation (follow MDPI’s guidelines) style in the main text.

Author Response

Reviewer 1

We thank the reviewer for the deep analysis of the manuscript. We think that the revisions suggested are appropriate and we tried to improve the manuscript according to that suggestions.

Major points:

  1. The authors should include more recent update on this topic and compare how this study further advances the current knowledge in the “Introduction section”.

We completly revised the manuscript following your suggestions, the “Introduction” was better organized and main concepts better described  Please see attached revised manuscript.

  1. Resolution of the bar graphics may be improved for the shake of clarity. Specially in Figures 2,6 and 7. And they should also be resized if appropriate.

In suggested figures (2,6 and 7) , the bar graphics were improved and we made corrections but we wanted to mantain the original graphs created by GraphPad program. Please see attached revised manuscript

  1. The Methods section in the study should be more accurately described for each technique used.

Following the guidelines of the Journal, we think that the technique were enough described. We corrected few errors in the description of main techniques and also add the detail for each used products. Please see attached revised manuscript

Minor points:

1.Unify the style of the references in the References Section and add DOI in the cases it is possible. And use the same reference and citation (follow MDPI’s guidelines) style in the main text.

We carefully cecked the style of the references in the Reference section and made corrections.

We hope that explanations satisfied the requests and really thank the reviewer for her/ his suggestions.

Reviewer 2 Report

The article is well written. However, some minor corrections are necessary to improve the quality of the manuscript.

The length of the abstract should be reduced if possible.

Check the grammatical errors and try to simplify the long sentences.

Include the latest references in the field from the last 2 to 3 years.

The authors should clearly state the purpose of the manuscript. The authors are advised to indicate the importance of this work.

Author Response

Reviewer 2

We thank the reviewer for the deep analysis of the submitted manuscript. Your suggestions and revisions allowed us to improve the quality of manuscript

 The length of the abstract should be reduced if possible.

The abstract was reduced as much as possible and organized better. Please see the revised manuscript as attached file-

Check the grammatical errors and try to simplify the long sentences.

The manuscript was completly revised expecially in “Abstract”, “Introduction” and “Discussion “ sections to simplify long sentences. A full correction of grammatical errors has been done by expert. Please see the revised manuscript as attached file-

Include the latest references in the field from the last 2 to 3 years.

We revised the entire manuscript in order to insert latest references, but one limitations of the references of this topic is that “AP complex and intracellular trafficking” is a topic where the best review and papers has been written by authors not so recently. Anyhow we introduced a reference (Buser DP and Spang A-Cell and Developmental Biology 2023), the review summarize the main findings in the intracellular transport field. Please see the revised manuscript as attached file-

The authors should clearly state the purpose of the manuscript. The authors are advised to indicate the importance of this work.

The manuscript was completly revised expecially in “Abstract”, “Introduction” and “Discussion “ sections to simplify long sentences. A full correction of grammatical errors has been done by expert. Please see the revised manuscript as attached file-

We hope that explanations satisfied the requests and really thank the reviewer for her/ his suggestions.

Reviewer 3 Report

Comments and Suggestions for Authors

In the manuscript submitted by researcher and colleagues entitled "The Zebrafish ap1g1 is Indispensable for Fertility and Neuro- 2 development: A New Model to Understand Adaptinopathies ", several aspects should be considered:

- I recommend authors follow the guidelines for authors of the respective journal.

- The title of the manuscript should be include all the objectives of the study,it is not objective, and appealing.

- The abstract should be rewritten, since it is quite confusing, the information is not properly linked and the main objective of the work is unclear, the abstract should be divided, but without headings, into background, methods, results, and conclusion.  The use of abbreviations should be avoided.

Brief conclusion of the study should be supplied at the end of abstract part of abstract.

Details of literature supported with related and recent references  should be provided in the introduction

- They should pay attention throughout the document to the way they write. Standardize words in italics. and "m/z" should be in italics, units of measurement.

You should have more scientific rigor.

Briefly explain the purpose of this study at the end of the introduction.

Results

Interesting work and impressive results, but it needs to be arranged and presented better.

Although the results have been described very well it can be improved

Materials and methods

- Also pay attention to how they reference previous work. You should check the guidelines for the correct way to do this.

- The data on the equipment used should be complete, including information on the molecule, city, and country of production.

You have to be more specific.

The discussion portion can be improved

Conclusion

A conclusion can be more accurately explained

References

You should choose to reference more recent literature, preferably from the last decade.

Author Response

Reviewer 3

We thank the reviewer for deep analysis of the manuscript. We think that the revisions suggested are appropriate and we tried to improve the manuscript according to the suggestions.

In the manuscript submitted by researcher and colleagues entitled "The Zebrafish ap1g1 is Indispensable for Fertility and Neuro- 2 development: A New Model to Understand Adaptinopathies ", several aspects should be considered:

- I recommend authors follow the guidelines for authors of the respective journal.

- The title of the manuscript should be include all the objectives of the study,it is not objective, and appealing.

The title was changed and we proposed a new title “Deficency of AP1 complex ap1g1 in zebrafish model led to perturbation of neurodevelopment, female and male fertility; new insight to understand adaptinopathies”

- The abstract should be rewritten, since it is quite confusing, the information is not properly linked and the main objective of the work is unclear, the abstract should be divided, but without headings, into background, methods, results, and conclusion.  The use of abbreviations should be avoided.

Brief conclusion of the study should be supplied at the end of abstract part of abstract.

The abstract was reduced and organized better. Please see the revised manuscript as attached file-Line 18-37

Details of literature supported with related and recent references  should be provided in the introduction

We completely revised the manuscript and better described the introduction, we think that the choosed references are important for the style of the manuscript. Please see the revised manuscript

They should pay attention throughout the document to the way they write. Standardize words in italics. and "m/z" should be in italics, units of measurement.

You should have more scientific rigor.

Briefly explain the purpose of this study at the end of the introduction.

The manuscript was completly revised expecially in “Abstract”, “Introduction” and “Discussion “sections to simplify long sentences. A full correction of grammatical errors has been done by expert. Please see the revised manuscript as attached file-

Results

Interesting work and impressive results, but it needs to be arranged and presented better.

Although the results have been described very well it can be improved

The results has been revised and improved Please see the revised manuscript as attached file-

Materials and methods

The data on the equipment used should be complete, including information on the molecule, city, and country of production. (paper Dario)

You have to be more specific.

Following the guidelines of the Journal we revised the described techniques and described better. We corrected few errors in the description of main techniques and also add the detail for each used products. Please see attached revised manuscript

The discussion portion can be improved

Conclusion

A conclusion can be more accurately explained

Discussion has been completely revised- Please see attached revised manuscript Line 716-979

Conclusion has been added Line 977-985

References

You should choose to reference more recent literature, preferably from the last decade.

We carefully cecked the style of the references in the Reference section and made corrections.

We hope that explanations satisfied the requests and really thank the reviewer for her/ his suggestions.

Please see the attchment
